# *Cortinarius* and *Tomentella* Fungi Become Dominant Taxa in Taiga Soil after Fire Disturbance

**DOI:** 10.3390/jof9111113

**Published:** 2023-11-17

**Authors:** Zhichao Cheng, Song Wu, Hong Pan, Xinming Lu, Yongzhi Liu, Libin Yang

**Affiliations:** 1Key Laboratory of Biodiversity, Institute of Natural Resources and Ecology, Heilongjiang Academy of Sciences, Harbin 150040, China; chengzc928@163.com (Z.C.); panhong500@163.com (H.P.); luxinming0210@163.com (X.L.); 2Heilongjiang Huzhong National Nature Reserve, Daxing’anling 165038, China; 3Science and Technology Innovation Center, Institute of Scientifc and Technical Information of Heilongjiang Province, Harbin 150028, China; wusong0927@126.com

**Keywords:** soil fungi, taiga forests, fire sites, diversity, community structure

## Abstract

Fungi have important ecological functions in the soil of forests, where they decompose organic matter, provide plants with nutrients, increase plant water uptake, and improve plant resistance to adversity, disease, and disturbance. A forest fire presents a serious disturbance of the local ecosystem and can be considered an important component affecting the function of ecosystem biomes; however, the response of soil fungi to fire disturbance is largely unknown. To investigate the effects of fire disturbance on the community composition and diversity of soil fungi in a taiga forest, we collected soil from plots that had undergone a light, moderate, and heavy fire 10 years previously, with the inclusion of a fire-free control. The present soil fungi were characterized using Illumina MiSeq technology, and the sequences were analyzed to identify differences in the community composition and diversity in response to the changed soil physicochemical properties. The results showed that the Chao1 index, which characterizes the alpha diversity of the fungi, did not change significantly. In contrast, the Shannon index increased significantly (*p* < 0.05) and the Simpson index decreased significantly (*p* < 0.05) following a light or heavy fire disturbance compared to the control. The relative abundance of Basidiomycota was significantly higher in the soil of the fire sites than that in the control (*p* < 0.01), and the relative abundance of Ascomycota was significantly lower (*p* < 0.01). The results of principal coordinates analyses (PCoAs) showed that fire disturbance highly significantly affected the beta diversity of soil fungi (*p* < 0.001), while the results of canonical correlation analysis (CCA) indicated that the available nitrogen (AN), moisture content (MC), pH, available potassium (AK), and total nitrogen (TN) contents of the soil significantly affected the compositional structure and diversity of the soil fungal communities. The results of functional prediction showed that the majority of the detected soil fungi were symbiotrophs, followed by saprotrophs and saprotroph–symbiotrophs, with ectomycorrhiza being the dominant functional taxon. Fire disturbance significantly reduced the relative abundance of ectomycorrhiza (*p* < 0.05). This study illustrates that fire disturbance alters the structural composition, diversity, dominance, and relative abundance of the guilds of soil fungal communities in taiga forest, and strongly affected the beta diversity of soil fungi, with AN, MC, pH, AK, and TN being the most important factors affecting their community structure. The results may provide a useful reference for the restoration and rehabilitation of taiga forests after fire disturbance.

## 1. Introduction

A complex relationship with close interactions exists between forest fires and forest ecosystems [1]. On the one hand, the high temperatures generated by a fire can directly kill or burn fauna and flora [2]; on the other hand, fire can also have a significant impact on the structure and function of forest ecosystems by altering the composition of vegetation communities [3], the physicochemical properties of the soil [4] and the activities of soil microorganisms [5]; the latter have significant impacts on the structure and function of forest ecosystems. In recent years, multiple studies have shown that forest fires, although causing damage, can also have positive effects [6,7,8,9]. Fires can to some extent promote the ecological effectiveness of forest ecosystems [10], enable the succession and restoration of plant communities [11], promote soil nutrient cycling [12], stimulate the functional activity of soil microorganisms [13], and suppress pest outbreaks [14].

Fungi are important components of the microbial taxa present in soil [15,16] and function as decomposers in forest ecosystems. They participate in the cycling of organics [17], energy flow [18], and information transfer [19] within a complex network of relationships affected by soil physicochemical properties and the aboveground vegetation [20]. A forest fire can rapidly kill soil fungi [21] and the burning soil organic matter can result in a significant loss and depletion of soil nutrients that is detrimental to soil fungi [22]. At the same time, fungi can have positive effects on soils after a fire disturbance. Although the dynamic balance of the vegetation, soil, and soil microorganisms in a local ecosystem is immediately disturbed by a fire, the fungal mycelia can recombine the mineral particles into stable aggregates over time, which can stabilize the soil structure, regulating and restoring the plant–soil relationship, thus promoting forest recovery [23]. The effects of a forest fire on soil fungi are not uniformly described in the current literature. For instance, Meng et al. [24] found that the diversity and abundance of soil fungi decreased following light and moderate fires, with slight changes observed in the community structure; Day et al. [25] reported that the abundance and diversity of soil fungi decreased with increasing fire intensity, while Tian et al. [26] described that the fungal diversity in the 0–5 cm soil layer increased after a fire. Kutorga et al. [27] observed that the relative abundance of soil fungi returned to pre-fire levels two years after a fire. The reasons for these different findings may be related to the intensity of the fire, differences in the type of ecosystem under investigation, the heterogeneity of the local environment, and other factors. Although the scientific community has gained increasing understanding of the interactions and impacts of fire and soil fungi, the influencing factors are complex, and the pattern of changes and the mechanism of interference are still unclear.

The Greater Khingan mountains in Northeastern China are covered by typical taiga forests that are dominated by *Larix gmelinii*. This region, with a cold temperate climate, is frequently visited by fires, especially in spring and autumn when the air is dry, rainfall is low, and strong winds combine with a low water content in the local vegetation. This, together with a thick humus layer, makes taiga forests prone to forest wildfires. Research in this region has focused on the changes in soil physicochemical properties after fire disturbance [28], described the concentration of pyrolytic carbon produced by fires [29], investigated the transformation of trace metals [30], and reported the effects of fire disturbance on tree growth [31]. The effects of fire on the local soil fungal community composition and on soil physicochemical properties have not been thoroughly addressed. In this study, past (2010) fire sites with different fire intensities (light, moderate, and heavy fire) of taiga forest in the Huzhong National Nature Reserve, Greater Khingan area, Heilongjiang Province of China, were investigated. The soil fungal community characteristics of these various fire sites were analyzed using Illumina MiSeq technology. The collected data can eventually be used to enable activities for improving soil structure and restoring forest ecosystems after a fire disturbance.

## 2. Materials and Methods

### 2.1. Location of the Sample Plots

The study site is located in Huzhong National Nature Reserve, Heilongjiang Province, in China (122°12′16.3′′–122°21′7.8′′ E, 53°26′30.6′′–53°28′6.3′′ N, Figure 1) and is an area of ongoing investigations in our research group that was recently described in more detail [32]. The area has a cold-temperate continental monsoon climate with marked seasonal temperature differences, with a sudden rise in temperature in spring, a warm and short summer with abundant rainfall, a sudden fall in temperature in autumn, and a long, cold, and snowy winter. The average annual temperature is −4 °C, average annual rainfall is 458.3 mm, average annual relative humidity is 71%, and average annual evaporation is 911 mm. The frost-free period usually lasts 80–100 days [33]. This area is a national nature reserve and represents one of the best preserved and most typical cold-temperate coniferous forest ecosystems in China. The dominant vegetation consists of *Larix gmelinii*, *Betula platyphylla* and *Pinus sylvestris* var. *Mongolica*.

### 2.2. Soil Sample Collection

Collection plots measuring 20 m × 20 m were identified at a constant altitude that had undergone past fires of light (L), moderate (M) or heavy (H) intensity 10 years ago (Light intensity: fire disturbance results in the scorching or complete combustion of ground vegetation, yet the soil’s organic litter layer remains intact. Moderate intensity: fire disturbance leads to the complete scorching or burning of ground vegetation as well, but it also results in the incomplete combustion of the litter layer. Heavy intensity: fire disturbance extends beyond the vegetation and litter layer, completely burning out both and additionally scorching or burning the humus surface layer of the soil to a variable extent). A pristine control (CK) plot was included, as previously described [32]. Details of the sampling regime are also described in that publication. In brief, 3 sample locations were chosen along the diagonal of each sample plot, giving a total of 12 samples. Following the removal of the humus and apomictic layer on the soil surface, soil was collected with an auger (10 cm diameter) from the 0–10 cm soil layer. After the removal of coarse material and sieving through a 2 mm nylon sieve, the soil was partitioned and one part was air-dried at room temperature for the determination of soil physical and chemical properties as previously described [32], while the other was subjected to Illumina MiSeq for fungal diversity analysis.

### 2.3. The Extraction and Sequencing of Soil Fungal DNA

Total microbial DNA was extracted using the Dneasy Power Soil DNA Extraction Kit (Qiagen, Hilden, Germany). The ITS region of the fungal rRNA gene was amplified using primers ITS1F (5′-CTTGGTCATTTAGAGGAAGTAA-3′) and ITS2R (5′-GCTGCGTTCTTCATCGATGC-3′) with conditions of PCR amplification as follows: 94 °C for 10 min, followed by 30 cycles of 90 °C for 60 s, 55 °C for 60 s, 72 °C for 60 s, and final termination at 72 °C for 10 min. The reaction products were purified using the QIA Quick PCR Purification Kit (Qiagen, Hilden, Germany). Sequence analysis was performed on an Illumina MiSeq sequencer as previously described [34]. The raw sequences were uploaded to the Sequence Read Archive (SRA) database (sequence number: PRJNA1025120).

### 2.4. Sequence Data Analysis

First, the bipartite sequences in Fastq format were screened one by one using the sliding window method, and those sequences that passed the initial screening by Halo were pairwise joined according to the overlapping bases using Flash software (http://ccb.jhu.edu/software/FLASH, v1.2.7, accessed on 13 July 2023). Questionable sequences were identified by Qiime2 then checked by Usearch v11, and, when applicable, chimeras were removed using the UCHIME2 software. Btrim software was used to remove low-quality regions (Q < 20), reads with one or more ambiguous sequences (N), and short sequences (less than 150 bp in length). The resulting high-quality sequences were clustered at 97% similarity to obtain Amplicon Sequence Variants (ASVs) based on the fungal database of Unite 8.0/itself_fungi (Unite Release 8.0) for comparison and identification [35]. Using Qiime2 software, the sequences were flattened using the naive Bayes annotation method with the minimum number of sample sequences as the criterion, and the taxonomic information of the species corresponding to each ASV was obtained.

Alpha diversity indices were calculated using mothur-1.30 software (https://www.Mothur.org/wiki/, accessed on 13 July 2023). The Chao1 index was calculated to compare the number of community species, and the Shannon and Simpson indices to compare the diversity and dominance of fungal community species [36]. Beta diversity was calculated based on Bray–Curtis distance and analyzed via principal coordinates analysis (PCoA). A canonical correlation analysis (CCA) and correlation heatmap analysis were performed using Pearson’s method for correlation analysis of environmental factors, as previously described [37]. Mapping was performed using the R 3.6.0 language vegan package and the ggplot2 package. Analysis of variance (ANOVA), multiple comparisons (Duncan′s test) and correlation analysis of different treatments were used as statistical tests, and these were performed using SPSS 25.0 software [38]. Potential ecological functions of soil fungi were analyzed using FUNGuild v1.0 software.

## 3. Results

### 3.1. The Physicochemical Properties of the Soil Are Affected by Past Fires

The physicochemical characteristics of the soil of the selected plots that had undergone light, moderate, heavy, or no fire have been described previously [32] and are briefly summarized as follows: The soil of all three fire sites had a significantly higher moisture content (MC) and contained higher amounts of microbial carbon (MBC) but less available potassium (AK) compared to CK. The nitrogen content, both in the form of available N (AN) and total N (TN), was higher in soil from the heavy-fire site than in soil from the other sites, while its pH was lower. The content of soil organic carbon (SOC) decreased with the increase in the past fire intensity.

### 3.2. Soil Fungal Diversity Varies between the Fire Sites

Following the sequencing of the ITS region, the diversity indices of the detected fungal communities were calculated. The Chao1 index of the soil fungi of all three fire sites was similar to that of the control (Table 1). The Shannon index was significantly higher (*p* < 0.05) and the Simpson index was significantly lower (*p* < 0.05) following light- and heavy-fire disturbance, though these indices did not change following moderate-fire disturbance (*p* > 0.05) (Table 1). This suggests that the number of soil fungal species did not change significantly after fire disturbance, but light- and heavy-fire disturbance increased the diversity and dominance of particular soil fungi, while moderate-fire disturbance more than likely resulted in a restoration of the soil fungi to pre-fire levels.

The soil fungal community composition of the soil samples was analyzed via principal coordinates analysis (PCoA) based on the Bray–Curtis distance. As shown in Figure 2, there was a significant difference in the soil fungal community structure between the different plots (Adonis analysis: R = 1, *p* = 0.001). The explanatory rate of principal component 1 was 39.96% and that of principal component 2 was 32.62%, giving a cumulative explanatory rate of 72.58%. Among the different sample types, the three CK samples were closely grouped together in the first quadrant, while the L and M groups were positioned in the second quadrant, and all these clustered in the positive direction of PC2. In contrast, the H group was found in the negative direction of PC2 and plotted in the fourth quadrant. This suggests that a fire disturbance 10 years previously had significantly altered the beta diversity of the local soil fungal communities in these taiga forests, and that the resultant soil fungal compositions were more similar between light- and moderate-fire disturbances but differed strongly in the control plot or following heavy fire.

### 3.3. The Structural Composition of Soil Fungal Communities at the Phylum and Genus Levels

The obtained sequences (34,554 valid reads) were attributed to fungal ASVs and their distribution is shown in a Venn diagram in Figure 3. In total, 1348 soil fungal ASVs were identified. The soil of H contained 544 ASVs, which was higher than that of M (316), L (442), or the fire-free control (420). Only 18 ASVs were shared by all soil types, and a relatively large number of ASVs were unique for each sample type. In particular, CK contained 326 unique ASVs that represented 77% of the total number of ASVs in that soil. For L, 236 or 53% unique ASVs were determined, 143 (45%) unique ASVs were found in M, and in H, 386 unique ASVs represented 71% of the total ASVs detected in that soil. Thus, the fraction of unique ASVs was highest in CK, followed by H, with the lowest in M, while the total number of ASVs was highest in H and again lowest in M.

The distribution of phyla is shown in Figure 4a. The dominant phylum of all soil types was Basidiomycota, representing 68.3% (in CK) to 52.67% (in L), followed by Ascomycota at 45.11% (in L) to 26.7% (in CK). Compared with the control soil, the relative abundance (r.a.) of Basidiomycota was significantly (*p* < 0.01) lower in all three fire sites and lower in L and M compared to H (*p* < 0.01). The relative abundance of Ascomycota was significantly (*p* < 0.01) higher in the soil disturbed by light or moderate fire compared to heavy fire, whereas the latter did not differ from the control (*p* > 0.05). There was a relatively large fraction of unclassified_k_fungi in the soil of H, which was significantly larger than in the other three soil types (*p* < 0.001). The higher fraction of Mortierellomycota in M was also significant (*p* < 0.05).

The findings were also analyzed at the genus level, and this identified vast differences between the soil types (Figure 4b). Whereas the control soil was strongly dominated by the Basidiomycota genus *Piloderma* (present at an r.a. of 59.5%), it was found at much lower levels in the fire sites. The dominant genus in M was *Cortinarius* (where it reached 30.32%), and with 15.19%, this Basidiomycota genus was also quite common in L. The soil of the M and L sites was relatively enriched in unclassified members of the family Hyalosyphaceae (Ascomycota), which was found at much lower levels in CK and H. In the soil of H, there was no truly dominant genus present: *Tomentella* (a Basidiomycota) was found at the highest r.a., but it only reached 10.87%. Compared with the CK soil, the r.a. of *Piloderma* was strongly significantly lower (*p* < 0.001), and that of *Cortinarius* was significantly higher (*p* < 0.01) in all fire sites. Enrichments of *Serendipita* in H and L, of *Russula* in M and H, and of *Thelephora*, *Oidiodendron,* and *Cenococcum* in M at significances of at least *p* < 0.01 were also noted. Overall, the results indicated that the dominance of fungal taxa present in the soil had changed as a result of a past fire, with a general trend of dominance shifting from *Piloderma* to *Cortinarius*, but affecting other genera as well. Thus, a fire had long-lasting effects on the composition of the soil fungi community at the genus level.

### 3.4. Correlation Analysis of Factors Influencing the Structure and Diversity of the Fungal Communities in the Soil Types

A correlation analysis was performed between the fungal alpha diversity indices (Shannon and Simpson) and the soil physicochemical properties, with the results shown in Table 2. The Shannon index significantly negatively correlated with AK (*p* < 0.05), significantly positively correlated with AP and TN (*p* < 0.05), and highly significantly positively correlated with moisture content MC (*p* < 0.001). The Simpson index significantly positively correlated with AK (*p* < 0.05) and significantly negatively correlated with MC (*p* < 0.05). Thus, MC, TN, AK and AP affected the diversity of the soil fungi, while MC and AK affected the degree of dominance of the community members.

Canonical correlation analysis (CCA) was carried out to identify correlations between the soil physicochemical properties and the community composition of the soil fungi. This analysis was performed at the ASV level, with the results plotted in Figure 5. The first two CCA axes explained 29.44% and 26.75% of the variance, respectively. The fungal communities in soil from L and M positively correlated with AP, SOC, and MBC, while that in H positively correlated with AN, MC, and TN. The statistical significance tests (Table 3) identified that soil AN, MC, pH, AK, and TN were the factors that had highly significant effects (*p* < 0.01, for MC *p* < 0.001) on the composition of the fungal communities in the soil.

Those genera with significant differences in relative abundance relating to the intensity of the past fire were selected and a numerical matrix was obtained for their Pearson’s correlation coefficients with soil physicochemical properties. The results are plotted in the heatmap shown in Figure 6. *Piloderma* strongly and positively correlated with AK (*p* < 0.001) and negatively with MC (*p* < 0.01) and AP (*p* < 0.05). *Cortinarius* strongly positively correlated with MBC (*p* < 0.01) and negatively with AK (*p* < 0.05). Serendipita even more strongly correlated positively with MBC (*p* < 0.001). *Tomentella* strongly and negatively correlated with pH (*p* < 0.001) and positively with AN, TN, and MC (*p* < 0.001). Correlations with less abundant genera were also identified (Figure 5). The pH always correlated negatively, while TN and MBC always correlated positively for those cases where significance was reached. It is evident that the abundance of fungal genera was affected by the listed soil physicochemical properties, with variation in the degree of correlation between the different genera.

### 3.5. Predictive Analyses of Soil Fungal Function in Fire Sites

The functions of the identified fungi were classified according to the FUNGuild functional classification (Appendix A) as symbiotroph, pathotroph, saprotroph or combinations thereof. Symbiotrophic fungi represented the most abundant trophic type in all soil types, followed by saprotrophs and saprotroph–symbiotrophs. In the guilds classification, there were 17 functional groups with a relative abundance exceeding 1%. Ectomycorrhiza represented the dominant functional group. Except for plant pathogens, plant pathogen–undefined saprotrophs and soil saprotrophs, all functional groups were significantly different (*p* < 0.05) between the fire sites and the control. Among them, the abundance of ectomycorrhiza,, undefined saprotroph–undefined biotrophs, and fungal Parasite-undefined saprotroph was significantly lower in the soil of the fire sites compared with the control (*p* < 0.05). The abundance of orchid mycorrhiza was significantly higher in the fire sites compared to the control (*p* < 0.05). These results indicated that fire disturbance significantly altered the relative abundance of soil fungi guild in the investigated taiga forests.

## 4. Discussion

### 4.1. The Effects of Fire Disturbance on Soil Physicochemical Properties

Forest fire disturbance mainly causes changes in soil physicochemical properties, such as soil water content, pH and soil nutrients, through the burning of organic matter. Fire disturbance in this study caused significant changes in soil MC, pH, AK, AN and TN. Among them, soil pH was significantly lower after heavy-fire disturbance compared with the unfired group, which was consistent with the previous results [39,40], which may be attributed to the altered microclimatic conditions such as soil water, heat, and air, accompanied by light, which accelerated the decomposition of the surface humus, and the mineral element ions produced adsorbed on the surface of soil colloid with the hydrogen ions adsorbed on the surface of soil colloids were exchanged into the soil solution, resulting in a decrease in soil pH [41].

The soil water content of the fire sites in this study was significantly higher than that of the unfired stands, which is consistent with the findings of Zhang et al. [42]. This may be due to the large-scale fall of trees after a fire and the decomposed fallen wood increasing the water absorption and water retention of the soil, resulting in an increase in soil water content [43].

The significant decrease in soil AK content in this study is consistent with the findings of Mcintosh et al. [44,45], which can be attributed to the production of large amounts of inorganic potassium carbonate after forest fires and the loss of highly water-soluble potassium carbonate with rainwater leaching, and secondly, the use of potassium by plants during their growth and recovery, the reason for the significant decrease in AK [46,47].

In this study, the AN and TN contents of heavily burned soils were significantly higher than those of unfired soils, which is consistent with previous results [48,49]. This may be due to the combined effect of inputs from surface vegetation and the leaching of burned litter, and accelerated nutrient mineralization due to high temperatures and high heat [50].

### 4.2. The Effects of Fire Disturbance on Fungal Alpha Diversity

Changes in soil physicochemical properties caused by fire disturbances may lead to the disappearance of or a reduction in certain fungal species in the soil, and this affects the alpha diversity of soil fungi [51]. In some studies, MC, AP, AK and TN were found to correlate with changes in the alpha diversity of soil fungi [52,53]. In our study, the differences in the soil physical and chemical properties of the fire sites did not change the total number of soil fungal species in this study, which is in line with the results of others [51]. As the aboveground vegetation and soil nutrients gradually recovered over time after the fire, the soil fungal communities were gradually restored to the original state. In this study, the species diversity and dominance of soil fungi increased significantly and were positively correlated with MC and AP, and negatively correlated with AK, which is consistent with the results of the previous works [54,55,56].

Several explanations for these observations can be postulated: First, the significant increase in diversity and dominance may be related to the increase in soil effective nutrients after light- and heavy-fire disturbances, which enhances the ability of soil fungi to grow and compete for external environmental resources, thus maintaining or even increasing soil fungal diversity and dominance [57]; the number of species, diversity and dominance of soil fungi under moderate-fire disturbance were not significantly different from those of the unfired group, which may be due to the inconsistent degree of the effect of different intensities of fire on soil physicochemical properties (AN, pH, TN, SOC and AP). Among them, the moderate-fire disturbance was more similar to the unfired sample plots, where new ecological niches could be created under such disturbance conditions, providing new resources and opportunities, which would enable species to better adapt to the environment and return them to a relatively stable state [58]. Second, an increase in soil MC would have increased plant productivity and SOC accumulation, which promotes soil fungal metabolism and reproduction, ultimately increasing soil fungal diversity and dominance [59,60]. Third, due to the uptake and use of soil AP by plant roots and its mycorrhizal symbiosis, soil fungi absorbed nutrients through apoplastic decomposition and root secretion, which enhanced fungal adaptability and symbiosis, which is consistent with previous findings [61,62]. In contrast, excessive AK levels inhibit enzymatic reaction processes, membrane homeostasis and osmotic pressure levels of soil microorganisms and suppress the growth and metabolism of soil fungi [63], which explains why increased AK levels reduced the alpha diversity of the soil fungal community.

### 4.3. The Effects of Fire Disturbance on Soil Fungal Community Composition

The high temperature and heat generated by a fire would have killed some soil fungi, after which the soil fungal community reassembled to form a new fungal community structure [21]. We observed an increase in the relative abundance of Ascomycota, combined with a decrease in Basidiomycota, in the fire sites compared to the fire-free control soil. That observation is consistent with previously published work [64]. It may be related to the function of Ascomycota in degrading hard-to-decompose organic matter such as cellulose and lignin [65]. Following a fire, there would be a large increase in the amount of dead leaves, other debris and ash accumulating on the soil surface, which would allow Ascomycota to fully exploit and decompose the hard-to-decompose organic matter [66], resulting in its dominance. On the other hand, Basidiomycota members are more sensitive to environmental perturbations than Ascomycetes [67]. Although they also have the ability to degrade lignin [68,69], they often form symbiotic relationships with plant roots, and the death or damage of trees and their root systems as a result of fire would also destroy this symbiosis, thus reducing the relative abundance of Basidiomycota.

We report that the genera *Cortinarius* and *Tomentella* became dominant in the fire sites, probably because *Cortinarius* and *Tomentella* play a protective and promoting role for growth and development in extreme environments [70]. Fire-adapted thermophilic fungi can replace fire-sensitive fungi in sites that are fire-prone, and their adaptability to thermal disturbance and ability to fully exploit environmental resources allow a rapid occupation of freed ecological niches after a fire disturbance, allowing them to grow into dominant populations [71].

Fire disturbance significantly altered the beta diversity of the soil fungi detected in this study, with significant variability in the community composition related to the past fire intensity, which is consistent with previous studies [72,73]. The genera *Cortinarius* and *Tomentella* were among the significantly different fungal dominant taxa in the fire sites, and the results of correlation analyses showed that *Cortinarius* correlated with AK and MBC and *Tomentella* strongly correlated with AN, MC, pH, and TN. These soil characteristics may represent the main influencing factors causing significant changes in soil beta diversity. Among them, nutrient contents such as N are important factors influencing the structural composition of microbial communities, which has been confirmed in studies on fungal associations in *Larix gmelinii* of the Greater Khingan Mountains [33,74]. Forest soil moisture content is an important factor in regulating soil microbial metabolism and in our study correlated positively to five abundant genera and negatively to three others. This illustrates that MC can affect soil microbial populations and their activities. The soil’s pH is also an important environmental factor affecting soil microbial communities and correlated negatively with the abundance of five genera. Thus, MC and pH are also important in regulating changes in fungal communities [74].

### 4.4. The Effects of Fire Disturbance on Potential Functional Taxa of Soil Fungi

As typical symbiotic fungi, ectomycorrhizal fungi promote the growth of vegetation through symbiosis with their plant hosts that provides nutrients needed for their own growth, and at the same time, improve the host’s ability to absorb nutrients [75]. The dominant genera of the Basidiomycota phylum detected in this study were mostly ectomycorrhizal fungi, which are considered to be fire-sensitive taxa [55,76,77]. Changes in ectomycorrhizal fungal communities after a fire will be influenced by plant colonization during subsequent restoration of the vegetation. This is consistent with the results reported here, which showed that the abundance of ectomycorrhizal fungi was significantly reduced after a fire. Thus, fire may affect the nutrient patterns of soil fungal communities but does not change the restoration strategy.

Che et al. [78] concluded that saprotrophic fungi, as important decomposers of soil organic matter, can provide more adequate nutrients to plant roots. In this study, we found that light-to-moderate fire disturbance increased the relative abundance of saprophytic fungi, indicating that such a fire disturbance is beneficial to the decomposition ability of the soil fungi. Meanwhile, this study also detected saprophytic–symbiotic fungi, and a large number of fungi with compatible functions were also present in the guild, which further indicates the diversity of fungal functions in the fire sites [79]. However, it should be noted that the relative abundance of plant pathogenic fungi in the fire group was higher than that in the control, and although the difference was not significant, it indicates that a past fire may increase the subsequent proportion of plant pathogenic fungi in the soil, which is consistent with the results of previous studies [80,81]. This increases the likelihood of a fungal infestation of vegetation in past fire sites. Therefore, we believe that fire not only accelerates the decomposition of soil organic matter but also increases the susceptibility of plants to disease, which ultimately affects the ecological stability of forests visited by fires.

## 5. Conclusions

(1) The number of species of soil fungi and their diversity did not change significantly following a moderate fire disturbance, but light and heavy fire disturbances significantly increased the diversity and dominance of soil fungi.

(2) Compared to the fire-free control soil, the relative abundance of Ascomycota was significantly increased, and that of Basidiomycota was significantly decreased in the fire sites.

(3) Fire disturbance significantly altered the beta diversity of soil fungi in the investigated taiga forest, and soil AN, MC, pH, AK and TN were the important factors influencing the changes in soil fungal community structure.

(4) The main trophic modes of the detected soil fungi were symbiotic, saprotrophic and saprotrophic–symbiotic, and ectomycorrhiza was the most dominant functional taxon, but fire disturbance significantly reduced the relative abundance of ectomycorrhiza.

## Figures and Tables

**Figure 1 jof-09-01113-f001:**
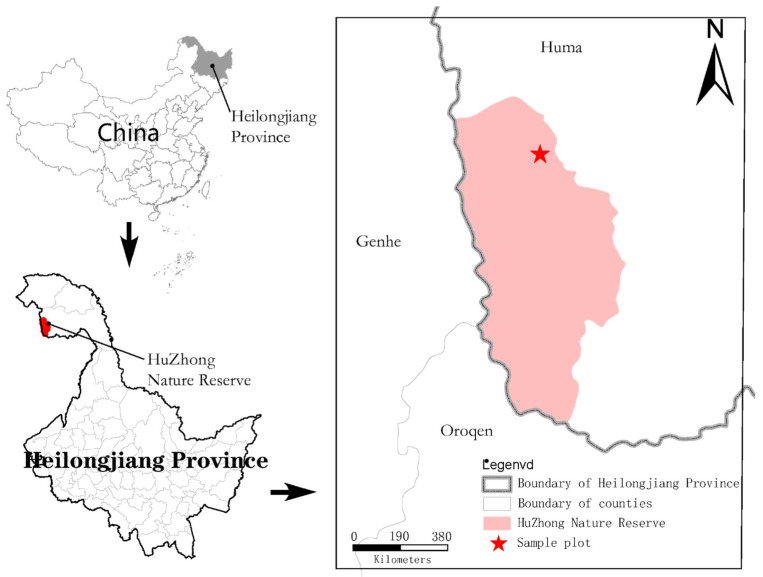
The asterisk indicates the study site in Heilongjiang Province and China.

**Figure 2 jof-09-01113-f002:**
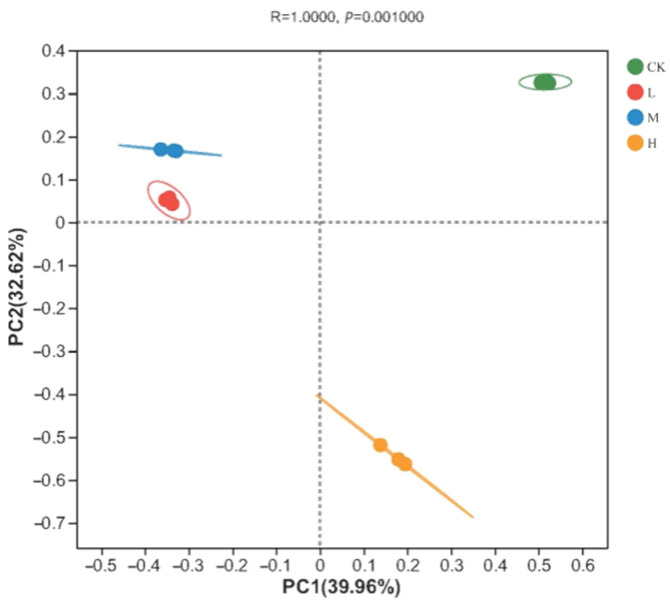
PCoA analysis of fungal communities in soil of different intensity fire sites with control (CK, no fire), light fire (L), moderate fire (M), and heavy fire (H). For each fire site, three independent samples were analyzed.

**Figure 3 jof-09-01113-f003:**
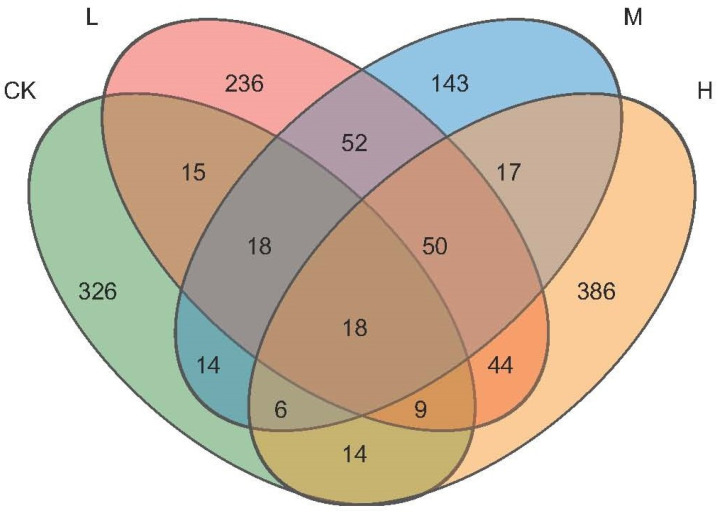
Venn diagram showing the distribution of soil fungi ASVs detected in the soil types.

**Figure 4 jof-09-01113-f004:**
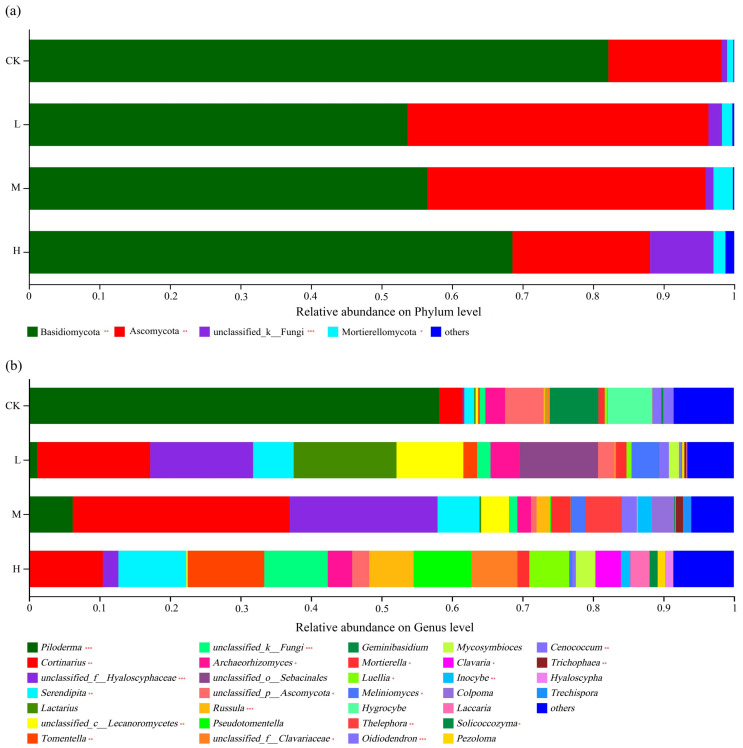
Soil fungal community compositions at the phylum (**a**) and genus (**b**) levels. Significance is indicated by red asterisks compared to CK with * *p* < 0.05, ** *p* < 0.01, *** *p* < 0.001.

**Figure 5 jof-09-01113-f005:**
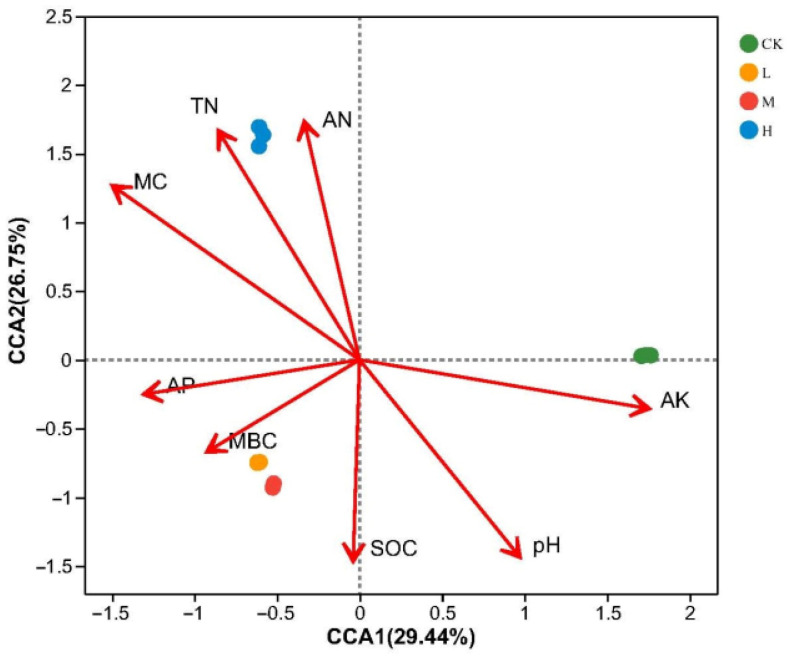
The canonical correspondence analysis (CCA) of soil fungal community and soil chemical and physical properties.

**Figure 6 jof-09-01113-f006:**
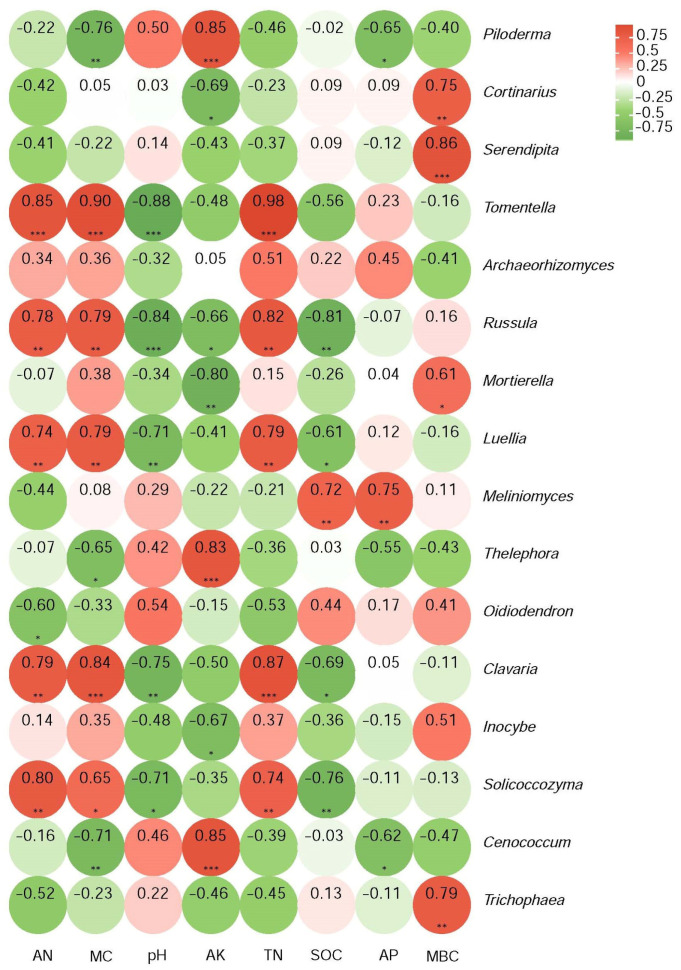
Pearson’s rank correlations between the relative abundances of genera and soil physicochemical parameters. Red identifies positive and green identifies negative correlations, with darker colors for stronger correlations. Significance is indicated as * *p* < 0.05, ** *p* < 0.01, *** *p* < 0.001.

**Table 1 jof-09-01113-t001:** Alpha diversity of fungal communities in soil with different past fire intensity. Different letters within a row indicate significant differences (*p* < 0.05; ANOVA) among the different intensities of fire in this study.

Fire Intensity	Chao1	Shannon	Simpson
CK (no fire)	261 ± 52.37 ab	2.67 ± 0.69 b	0.23 ± 0.16 a
L (light fire)	288 ± 7.94 a	3.74 ± 0.11 a	0.05 ± 0.01 b
M (moderate fire)	206.67 ± 16.5 b	3.34 ± 0.1 ab	0.07 ± 0.01 ab
H (heavy fire)	313 ± 60.56 a	3.85 ± 0.17 a	0.04 ± 0.00 b

**Table 2 jof-09-01113-t002:** Correlation coefficients between soil chemical and physical properties and fungal diversity indices. AN: available nitrogen; MC: moisture content; AK: available potassium; TN: total nitrogen; SOC: soil organic carbon; AP: available phosphorus, MBC: microbial carbon. Significance is indicated as * *p*< 0.05, ** *p* < 0.01.

	AN	MC	pH	AK	TN	SOC	AP	MBC
Chao1	0.58 *	0.45	−0.55	0.04	0.64 *	0.01	0.41	−0.54
Shannon	0.36	0.75 **	−0.56	−0.63 *	0.59 *	0.04	0.65 *	0.09
Simpson	−0.23	−0.60 *	0.42	0.64 *	−0.40	−0.04	−0.53	−0.24

**Table 3 jof-09-01113-t003:** Significance tests between soil physicochemical properties and the fungal community structures. Significance is indicated as * *p* < 0.05, ** *p* < 0.01, *** *p* < 0.001.

	R^2^	*p*-Value
AN	0.7631	0.008 **
MC	0.9553	0.001 ***
pH	0.7395	0.004 **
AK	0.794	0.005 **
TN	0.8663	0.004 **
SOC	0.5195	0.043 *
AP	0.4316	0.075
MBC	0.3122	0.146

## Data Availability

Data are contained within the article.

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
