# Peer review of "Cortinarius and Tomentella Fungi Become Dominant Taxa in Taiga Soil after Fire Disturbance"

_jof, 2023, doi:10.3390/jof9111113_

Round 1
Reviewer 1 Report
Comments and Suggestions for Authors
This manuscript deals with the diversity and structure of fungal communities in the soil of a taiga ecosystem affected by light, moderate and heavy intensity fires. It is a well-designed research work, the molecular techniques used are correct and current to carry out this study. Data analysis are appropriate to understand the results obtained. In my opinion, the weakness of this research lies in its lack of novelty and the results obtained, which are as expected, do not contribute anything new to the knowledge of fungal succession after fire. In this sense, conclusions 2,3 and 5 (conclusion number 4 does not exist) are known from the numerous papers published on this research topic. Regarding conclusion number 1 (“The number of species of soil fungi and their diversity did not change significantly following a moderate disturbance and their diversity did not chance, but light and heavy fire disturbances significantly increased the diversity and dominance of soil fungi”), which is also discussed in the results, the authors do not give a possible explanation for this fact.
In lines 93-94 of the introduction (and in the Abstract), the authors highlight that the data obtained “can eventually be used to enable activities for improving soil structure and restoring forest ecosystems after a fire disturbance”; How could these results be applied to the restoration of ecosystems affected by fire? , It would be very interesting if the authors explained something about this statement since it is part of the objectives of this research.
Perhaps the research would be more novel and the results more relevant if the authors had included arbuscular mycorrhizal fungi in the study, which are a very important component of the soil fungal flora and are probably are forming symbiotic relationship with roots of the shrubs and herbaceous plants species under the tree layer and that play a fundamental role in recovery after a disturbance.
Author Response
Thank you to the experts for their valuable inputs, the responses to which are annexed.

Reviewer 2 Report
Comments and Suggestions for Authors
The manuscript “Cortinarius and Tomentella fungi become dominant taxa in taiga soil after fire disturbance” by Zhichao Cheng, Song Wu, Hong Pan, Xinming Lu, Yongzhi Liu and Libin Yang is dedicated to studying the effects of fire disturbance on the community composition and diversity of soil fungi in a taiga forest. The authors discuss important questions about the effects of fires on the diversity of soil fungal complexes as well as issues of the ecological stability of the fire-affected forest. The article is well written and fits the profile of the journal.
Minor comments
Line 26: Please, “which belonging” change to “which belong”
Line 55: “…..and assist in the reduction of pests and disease populations [14].” Please, explain, what does it mean, “disease populations” (?). I could not find the referenced article [14] to find the explanation.
Line 66: “….The effects a forest fire…” Please, change to “The effects of a forest fire…”
Introduction: Please add a few sentences explaining the differences between the fire intensities you mentioned (light, moderate and heavy fire). Although you characterize fires in your other paper [32] that you refer to in the methods, it would still be more convenient for the readers of this paper to have a brief introduction here as well.
Lines 217-218: “The representative sequences of the ASVs were analyzed taxonomically at a 97% similarity level, which identified a total of 12 phyla, 41 classes, 88 orders, 163 families, 218 genera and 398 species of fungi.” Have the species been identified? If so, the results of the fungal species identification should be presented and the data discussed. If not, this information is not relevant for this article. Please, clarify.
Line 259: “between soil environmental factors”, soil chemical and physical properties would be better.
Comments on the Quality of English LanguageMinor editing of English language required
Author Response
Thank you for your valuable feedback. We have made modifications based on your feedback, as detailed in the attachment.
